# Compete to Compute

**Rupesh Kumar Srivastava, Jonathan Masci, Sohrob Kazerounian,**
**Faustino Gomez, Jürgen Schmidhuber**
IDSIA, USI-SUPSI
Manno–Lugano, Switzerland
{rupesh, jonathan, sohrob, tino, juergen}@idsia.ch

## Abstract

Local competition among neighboring neurons is common in biological neural networks (NNs). In this paper, we apply the concept to gradient-based, backprop-trained artificial multilayer NNs. NNs with competing linear units tend to outperform those with non-competing nonlinear units, and avoid catastrophic forgetting when training sets change over time.

## 1 Introduction

Although it is often useful for machine learning methods to consider how nature has arrived at a particular solution, it is perhaps more instructive to first understand the functional role of such biological constraints. Indeed, artificial neural networks, which now represent the state-of-the-art in many pattern recognition tasks, not only resemble the brain in a superficial sense, but also draw on many of its computational and functional properties.

One of the long-studied properties of biological neural circuits which has yet to fully impact the machine learning community is the nature of local competition. That is, a common finding across brain regions is that neurons exhibit on-center, off-surround organization [1, 2, 3], and this organization has been argued to give rise to a number of interesting properties across networks of neurons, such as winner-take-all dynamics, automatic gain control, and noise suppression [4].

In this paper, we propose a biologically inspired mechanism for artificial neural networks that is based on local competition, and ultimately relies on local winner-take-all (LWTA) behavior. We demonstrate the benefit of LWTA across a number of different networks and pattern recognition tasks by showing that LWTA not only enables performance comparable to the state-of-the-art, but moreover, helps to prevent *catastrophic forgetting* [5, 6] common to artificial neural networks when they are first trained on a particular task, then abruptly trained on a new task. This property is desirable in continual learning wherein learning regimes are not clearly delineated [7]. Our experiments also show evidence that a type of modularity emerges in LWTA networks trained in a supervised setting, such that different modules (subnetworks) respond to different inputs. This is beneficial when learning from multimodal data distributions as compared to learning a monolithic model.

In the following, we first discuss some of the relevant neuroscience background motivating local competition, then show how we incorporate it into artificial neural networks, and how LWTA, as implemented here, compares to alternative methods. We then show how LWTA networks perform on a variety of tasks, and how it helps buffer against catastrophic forgetting.

## 2 Neuroscience Background

Competitive interactions between neurons and neural circuits have long played an important role in biological models of brain processes. This is largely due to early studies showing that

many cortical [3] and sub-cortical (e.g., hippocampal [1] and cerebellar [2]) regions of the brain exhibit a recurrent on-center, off-surround anatomy, where cells provide excitatory feedback to nearby cells, while scattering inhibitory signals over a broader range. Biological modeling has since tried to uncover the functional properties of this sort of organization, and its role in the behavioral success of animals.

The earliest models to describe the emergence of winner-take-all (WTA) behavior from local competition were based on Grossberg's shunting short-term memory equations [4], which showed that a center-surround structure not only enables WTA dynamics, but also contrast enhancement, and normalization. Analysis of their dynamics showed that networks with slower-than-linear signal functions uniformize input patterns; linear signal functions preserve and normalize input patterns; and faster-than-linear signal functions enable WTA dynamics. Sigmoidal signal functions which contain slower-than-linear, linear, and faster-than-linear regions enable the supression of noise in input patterns, while contrast-enhancing, normalizing and storing the relevant portions of an input pattern (a form of soft WTA). The functional properties of competitive interactions have been further studied to show, among other things, the effects of distance-dependent kernels [8], inhibitory time lags [8, 9], development of self-organizing maps [10, 11, 12], and the role of WTA networks in attention [13]. Biological models have also been extended to show how competitive interactions in spiking neural networks give rise to (soft) WTA dynamics [14], as well as how they may be efficiently constructed in VLSI [15, 16].

Although competitive interactions, and WTA dynamics have been studied extensively in the biological literature, it is only more recently that they have been considered from computational or machine learning perspectives. For example, Maas [17, 18] showed that feedforward neural networks with WTA dynamics as the only non-linearity are as computationally powerful as networks with threshold or sigmoidal gates; and, networks employing only soft WTA competition are universal function approximators. Moreover, these results hold, even when the network weights are strictly positive—a finding which has ramifications for our understanding of biological neural circuits, as well as the development of neural networks for pattern recognition. The large body of evidence supporting the advantages of locally competitive interactions makes it noteworthy that this simple mechanism has not provoked more study by the machine learning community. Nonetheless, networks employing local competition have existed since the late 80s [21], and, along with [22], serve as a primary inspiration for the present work. More recently, *maxout* networks [19] have leveraged locally competitive interactions in combination with a technique known as *dropout* [20] to obtain the best results on certain benchmark problems.

## 3 Networks with local winner-take-all blocks

This section describes the general network architecture with locally competing neurons. The network consists of *B blocks* which are organized into layers (Figure 1). Each block, $b_i, i = 1..B$, contains $n$ computational units (neurons), and produces an output vector $\mathbf{y}_i$, determined by the local interactions between the individual neuron activations in the block:

$$y_i^j = g(h_i^1, h_i^2..., h_i^n), \tag{1}$$

where $g(\cdot)$ is the *competition/interaction function*, encoding the effect of local interactions in each block, and $h_i^j, j = 1..n$, is the activation of the $j$-th neuron in block $i$ computed by:

$$h_i = f(\mathbf{w}_{ij}^T \mathbf{x}), \tag{2}$$

where $\mathbf{x}$ is the input vector from neurons in the previous layer, $\mathbf{w}_{ij}$ is the weight vector of neuron $j$ in block $i$, and $f(\cdot)$ is a (generally non-linear) activation function. The output activations $\mathbf{y}$ are passed as inputs to the next layer. In this paper we use the winner-take-all interaction function, inspired by studies in computational neuroscience. In particular, we use the hard winner-take-all function:

$$y_i^j = \begin{cases} h_i^j & \text{if } h_i^j \geq h_i^k, \ \forall k = 1..n \\ 0 & \text{otherwise.} \end{cases}$$

In the case of multiple winners, ties are broken by index precedence. In order to investigate the capabilities of the hard winner-take-all interaction function in isolation, $f(x) = x$

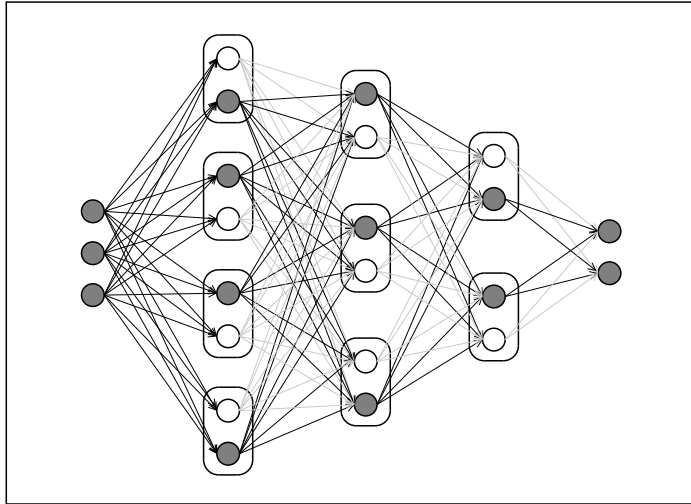

Figure 1: A Local Winner-Take-All (LWTA) network with blocks of size two showing the winning neuron in each block (shaded) for a given input example. Activations flow forward only through the winning neurons, errors are backpropagated through the active neurons. Greyed out connections do not propagate activations. The active neurons form a subnetwork of the full network which changes depending on the inputs.

(identity) is used for the activation function in equation (2). The difference between this Local Winner Take All (LWTA) network and a standard multilayer perceptron is that no non-linear activation functions are used, and during the forward propagation of inputs, local competition between the neurons in each block *turns off* the activation of all neurons except the one with the highest activation. During training the error signal is only backpropagated through the winning neurons.

In a LWTA layer, there are as many neurons as there are blocks active at any one time for a given input pattern[1]. We denote a layer with blocks of size $n$ as LWTA-$n$. For each input pattern presented to a network, only a subgraph of the full network is active, e.g. the highlighted neurons and synapses in figure 1. Training on a dataset consists of simultaneously training an exponential number of models that share parameters, as well as learning which model should be active for each pattern. Unlike networks with sigmoidal units, where all of the free parameters need to be set properly for all input patterns, only a subset is used for any given input, so that patterns coming from very different sub-distributions can potentially be modelled more efficiently through specialization. This modular property is similar to that of networks with rectified linear units (ReLU) which have recently been shown to be very good at several learning tasks (links with ReLU are discussed in section 4.3).

## 4   Comparison with related methods

### 4.1   Max-pooling

Neural networks with max-pooling layers [23] have been found to be very useful, especially for image classification tasks where they have achieved state-of-the-art performance [24, 25]. These layers are usually used in convolutional neural networks to subsample the representation obtained after convolving the input with a learned filter, by dividing the representation into pools and selecting the maximum in each one. Max-pooling lowers the computational burden by reducing the number of connections in subsequent convolutional layers, and adds translational/rotational invariance.

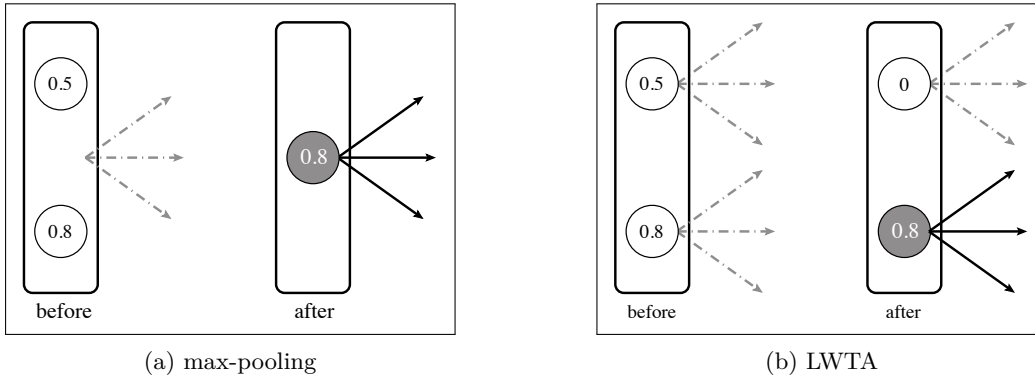

(a) max-pooling            (b) LWTA

Figure 2: Max-pooling vs. LWTA. (a) In max-pooling, each group of neurons in a layer has a single set of output weights that transmits the winning unit's activation (0.8 in this case) to the next layer, i.e. the layer activations are subsampled. (b) In an LWTA block, there is no subsampling. The activations flow into subsequent units via a different set of connections depending on the winning unit.

At first glance, the max-pooling seems very similar to a WTA operation, however, the two differ substantially: there is no downsampling in a WTA operation and thus the number of features is not reduced, instead the representation is "sparsified" (see figure 2).

## 4.2 Dropout

Dropout [20] can be interpreted as a model-averaging technique that jointly trains several models sharing subsets of parameters and input dimensions, or as data augmentation when applied to the input layer [19, 20]. This is achieved by probabilistically omitting ("dropping") units from a network for each example during training, so that those neurons do not participate in forward/backward propagation. Consider, hypothetically, training an LWTA network with blocks of size two, and selecting the winner in each block at random. This is similar to training a neural network with a dropout probability of 0.5. Nonetheless, the two are fundamentally different. Dropout is a regularization technique while in LWTA the interaction between neurons in a block replaces the per-neuron non-linear activation.

Dropout is believed to improve generalization performance since it forces the units to learn independent features, without relying on other units being active. During testing, when propagating an input through the network, all units in a layer trained with dropout are used with their output weights suitably scaled. In an LWTA network, no output scaling is required. A fraction of the units will be inactive for each input pattern depending on their total inputs. Viewed this way, WTA is restrictive in that only a fraction of the parameters are utilized for each input pattern. However, we hypothesize that the freedom to use different subsets of parameters for different inputs allows the architecture to learn from multimodal data distributions more accurately.

## 4.3 Rectified Linear units

Rectified Linear Units (ReLU) are simply linear neurons that clamp negative activations to zero ($f(x) = x$ if $x > 0$, $f(x) = 0$ otherwise). ReLU networks were shown to be useful for Restricted Boltzmann Machines [26], outperformed sigmoidal activation functions in deep neural networks [27], and have been used to obtain the best results on several benchmark problems across multiple domains [24, 28].

Consider an LWTA block with two neurons compared to two ReLU neurons, where $x_1$ and $x_2$ are the weighted sum of the inputs to each neuron. Table 1 shows the outputs $y_1$ and $y_2$ in all combinations of positive and negative $x_1$ and $x_2$, for ReLU and LWTA neurons. For both ReLU and LWTA neurons, $x_1$ and $x_2$ are passed through as output in half of the possible cases. The difference is that in LWTA both neurons are never active or inactive at the same time, and the activations and errors flow through exactly one neuron in the block. For ReLU neurons, being inactive (saturation) is a potential drawback since neurons that

Table 1: Comparison of rectified linear activation and LWTA-2.

| | | ReLU neurons | | LWTA neurons | |
|---|---|---|---|---|---|
| $x_1$ | $x_2$ | $y_1$ | $y_2$ | $y_1$ | $y_2$ |
| $x_1 > x_2$ | | | | | |
| Positive | Positive | $x_1$ | $x_2$ | $x_1$ | 0 |
| Positive | Negative | $x_1$ | 0 | $x_1$ | 0 |
| Negative | Negative | 0 | 0 | $x_1$ | 0 |
| $x_2 > x_1$ | | | | | |
| Positive | Positive | $x_1$ | $x_2$ | 0 | $x_2$ |
| Negative | Positive | 0 | $x_2$ | 0 | $x_2$ |
| Negative | Negative | 0 | 0 | 0 | $x_2$ |

do not get activated will not get trained, leading to wasted capacity. However, previous work suggests that there is no negative impact on optimization, leading to the hypothesis that such hard saturation helps in credit assignment, and, as long as errors flow through certain paths, optimization is not affected adversely [27]. Continued research along these lines validates this hypothesis [29], but it is expected that it is possible to train ReLU networks better.

While many of the above arguments for and against ReLU networks apply to LWTA networks, there is a notable difference. During training of an LWTA network, inactive neurons can become active due to training of the other neurons in the same block. This suggests that LWTA nets may be less sensitive to weight initialization, and a greater portion of the network's capacity may be utilized.

## 5 Experiments

In the following experiments, LWTA networks were tested on various supervised learning datasets, demonstrating their ability to learn useful internal representations without utilizing any other non-linearities. In order to clearly assess the utility of local competition, no special strategies such as augmenting data with transformations, noise or dropout were used. We also did not encourage sparse representations in the hidden layers by adding activation penalties to the objective function, a common technique also for ReLU units. Thus, our objective is to evaluate the value of using LWTA rather than achieving the absolute best testing scores. Blocks of size two are used in all the experiments.[2]

All networks were trained using stochastic gradient descent with mini-batches, learning rate $l_t$ and momentum $m_t$ at epoch $t$ given by

$$
\begin{aligned}
\alpha_t &= \begin{cases} \alpha_0 \lambda^t & \text{if } \alpha_t > \alpha_{min} \\ \alpha_{min} & \text{otherwise} \end{cases} \\
m_t &= \begin{cases} \frac{t}{T} m_i + (1 - \frac{t}{T}) m_f & \text{if } t < T \\ p_f & \text{if } t \geq T \end{cases}
\end{aligned}
$$

where $\lambda$ is the learning rate annealing factor, $\alpha_{min}$ is the lower learning rate limit, and momentum is scaled from $m_i$ to $m_f$ over $T$ epochs after which it remains constant at $m_f$. L2 weight decay was used for the convolutional network (section 5.2), and max-norm normalization for other experiments. This setup is similar to that of [20].

### 5.1 Permutation Invariant MNIST

The MNIST handwritten digit recognition task consists of 70,000 28x28 images (60,000 training, 10,000 test) of the 10 digits centered by their center of mass [33]. In the permutation invariant setting of this task, we attempted to classify the digits without utilizing the 2D structure of the images, e.g. every digit is a vector of pixels. The last 10,000 examples in the training set were used for hyperparameter tuning. The model with the best hyperparameter setting was trained until convergence on the full training set. Mini-batches of size 20 were

Table 2: Test set errors on the permutation invariant MNIST dataset for methods without data augmentation or unsupervised pre-training

| Activation | Test Error |
|---|---|
| Sigmoid [32] | 1.60% |
| ReLU [27] | 1.43% |
| ReLU + dropout in hidden layers [20] | 1.30% |
| **LWTA-2** | **1.28%** |

Table 3: Test set errors on MNIST dataset for convolutional architectures with no data augmentation. Results marked with an asterisk use layer-wise unsupervised feature learning to pre-train the network and global fine tuning.

| Architecture | Test Error |
|---|---|
| 2-layer CNN + 2 layer MLP [34] * | 0.60% |
| **2-layer ReLU CNN + 2 layer LWTA-2** | **0.57%** |
| 3-layer ReLU CNN [35] | 0.55% |
| 2-layer CNN + 2 layer MLP [36] * | 0.53% |
| 3-layer ReLU CNN + stochastic pooling [33] | 0.47% |
| 3-layer maxout + dropout [19] | 0.45% |

used, the pixel values were rescaled to $[0, 1]$ (no further preprocessing). The best model obtained, which gave a test set error of **1.28%**, consisted of three LWTA layers of 500 blocks followed by a 10-way softmax layer. To our knowledge, this is the best reported error, without utilizing implicit/explicit model averaging, for this setting which does not use deformations/noise to enhance the dataset or unsupervised pretraining. Table 2 compares our results with other methods which do not use unsupervised pre-training. The performance of LWTA is comparable to that of a ReLU network with dropout in the hidden layers. Using dropout in input layers as well, lower error rates of 1.1% using ReLU [20] and 0.94% using maxout [19] have been obtained.

## 5.2 Convolutional Network on MNIST

For this experiment, a convolutional network (CNN) was used consisting of $7 \times 7$ filters in the first layer followed by a second layer of $6 \times 6$, with 16 and 32 maps respectively, and ReLU activation. Every convolutional layer is followed by a $2 \times 2$ max-pooling operation. We then use two LWTA-2 layers each with 64 blocks and finally a 10-way softmax output layer. A weight decay of 0.05 was found to be beneficial to improve generalization. The results are summarized in Table 3 along with other state-of-the-art approaches which do not use data augmentation (for details of convolutional architectures, see [33]).

## 5.3 Amazon Sentiment Analysis

LWTA networks were tested on the Amazon sentiment analysis dataset [37] since ReLU units have been shown to perform well in this domain [27, 38]. We used the balanced subset of the dataset consisting of reviews of four categories of products: *Books, DVDs, Electronics* and *Kitchen appliances*. The task is to classify the reviews as positive or negative. The dataset consists of 1000 positive and 1000 negative reviews in each category. The text of each review was converted into a binary feature vector encoding the presence or absence of unigrams and bigrams. Following [27], the 5000 most frequent vocabulary entries were retained as features for classification. We then divided the data into 10 equal balanced folds, and tested our network with cross-validation, reporting the mean test error over all folds. ReLU activation was used on this dataset in the context of unsupervised learning with denoising autoencoders to obtain sparse feature representations which were used for classification. We trained an LWTA-2 network with three layers of 500 blocks each in a supervised setting to directly classify each review as positive or negative using a 2-way softmax output layer. We obtained mean accuracies of *Books*: 80%, *DVDs*: 81.05%, *Electronics*: 84.45% and *Kitchen*: 85.8%, giving a mean accuracy of **82.82%**, compared to 78.95% reported in [27] for denoising autoencoders using ReLU and unsupervised pre-training to find a good initialization.

Table 4: LWTA networks outperform sigmoid and ReLU activation in remembering dataset P1 after training on dataset P2.

| Testing error on P1 | LWTA | Sigmoid | ReLU |
|---|---|---|---|
| After training on P1 | $1.55 \pm 0.20\%$ | $1.38 \pm 0.06\%$ | $1.30 \pm 0.13\%$ |
| After training on P2 | $6.12 \pm 3.39\%$ | $57.84 \pm 1.13\%$ | $16.63 \pm 6.07\%$ |

## 6  Implicit long term memory

This section examines the effect of the LWTA architecture on catastrophic forgetting. That is, does the fact that the network implements multiple models allow it to retain information about dataset $A$, even after being trained on a different dataset $B$? To test for this *implicit long term memory*, the MNIST training and test sets were each divided into two parts, P1 containing only digits $\{0, 1, 2, 3, 4\}$, and P2 consisting of the remaining digits $\{5, 6, 7, 8, 9\}$. Three different network architectures were compared: (1) three LWTA layers each with 500 blocks of size 2, (2) three layers each with 1000 sigmoidal neurons, and (3) three layers each of 1000 ReLU neurons. All networks have a 5-way softmax output layer representing the probability of an example belonging to each of the five classes. All networks were initialized with the same parameters, and trained with a fixed learning rate and momentum.

Each network was first trained to reach a 0.03 log-likelihood error on the P1 training set. This value was chosen heuristically to produce low test set errors in reasonable time for all three network types. The weights for the output layer (corresponding to the softmax classifier) were then stored, and the network was trained further, starting with new initial random output layer weights, to reach the same log-likelihood value on P2. Finally, the output layer weights saved from P1 were restored, and the network was evaluated on the P1 test set. The experiment was repeated for 10 different initializations.

Table 4 shows that the LWTA network *remembers* what was learned from P1 much better than sigmoid and ReLU networks, though it is notable that the sigmoid network performs much worse than both LWTA and ReLU. While the test error values depend on the learning rate and momentum used, LWTA networks tended to remember better than the ReLU network by about a factor of two in most cases, and sigmoid networks always performed much worse. Although standard network architectures are known to suffer from catastrophic forgetting, we not only show here, for the first time, that ReLU networks are actually quite good in this regard, and moreover, that they are outperformed by LWTA. We expect this behavior to manifest itself in competitive models in general, and to become more pronounced with increasingly complex datasets. The neurons encoding specific features in one dataset are not affected much during training on another dataset, whereas neurons encoding common features can be reused. Thus, LWTA may be a step forward towards models that do not forget easily.

## 7  Analysis of subnetworks

A network with a single LWTA-m of $N$ blocks consists of $m^N$ subnetworks which can be selected and trained for individual examples while training over a dataset. After training, we expect the subnetworks consisting of active neurons for examples from the same class to have more neurons in common compared to subnetworks being activated for different classes. In the case of relatively simple datasets like MNIST, it is possible to examine the number of common neurons between *mean* subnetworks which are used for each class. To do this, which neurons were active in the layer for each example in a subset of 10,000 examples were recorded. For each class, the subnetwork consisting of neurons active for at least 90% of the examples was designated the representative mean subnetwork, which was then compared to all other class subnetworks by counting the number of neurons in common.

Figure 3a shows the fraction of neurons in common between the mean subnetworks of each pair of digits. Digits that are morphologically similar such as "3" and "8" have subnetworks with more neurons in common than the subnetworks for digits "1" and "2" or "1" and "5" which are intuitively less similar. To verify that this subnetwork specialization is a result of training, we looked at the fraction of common neurons between all pairs of digits for the

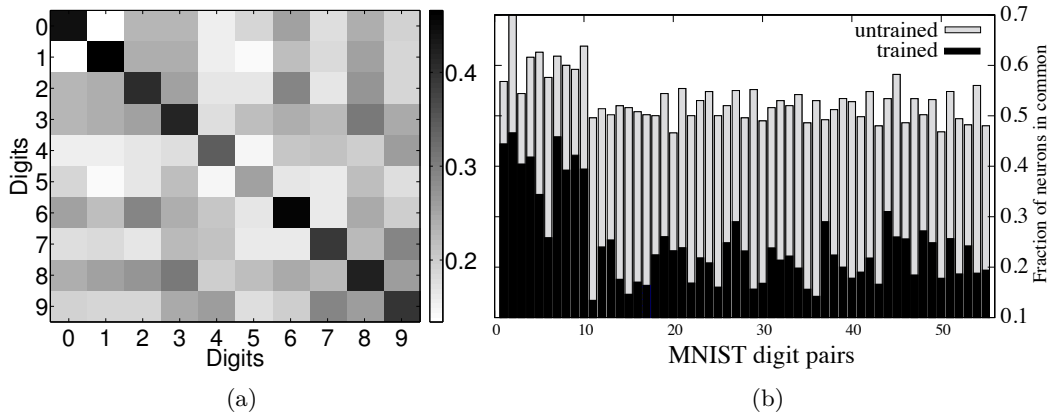

Figure 3: (a) Each entry in the matrix denotes the fraction of neurons that a pair of MNIST digits has in common, on average, in the subnetworks that are most active for each of the two digit classes. (b) The fraction of neurons in common in the subnetworks of each of the 55 possible digit pairs, before and after training.

same 10000 examples both before and after training (Figure 3b). Clearly, the subnetworks were much more similar prior to training, and the full network has learned to partition its parameters to reflect the structure of the data.

## 8   Conclusion and future research directions

Our LWTA networks automatically self-modularize into multiple parameter-sharing subnetworks responding to different input representations. Without significant degradation of state-of-the-art results on digit recognition and sentiment analysis, LWTA networks also avoid catastrophic forgetting, thus retaining useful representations of one set of inputs even after being trained to classify another. This has implications for continual learning agents that should not forget representations of parts of their environment when being exposed to other parts. We hope to explore many promising applications of these ideas in the future.

### Acknowledgments

This research was funded by EU projects WAY (FP7-ICT-288551), NeuralDynamics (FP7-ICT-270247), and NASCENCE (FP7-ICT-317662); additional funding from ArcelorMittal.

## Footnotes

[1]However, there is always the possibility that the winning neuron in a block has an activation of exactly zero, so that the block has no output.

[2]To speed up our experiments, the Gnumpy [30] and CUDAMat [31] libraries were used.

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
