[Reviews · NeurIPS 2013]

Submitted by Assigned_Reviewer_1

Winner-take-all modules are incorporated into an otherwise
feed-forward architecture, and trained via backpropagation. The
implementation is very straightforward, and the measured performance
is impressive.

We should not let the simplicity, and lack of associated mathematics
or analysis, preclude acceptance. This may be an important discovery,
and there are a variety of obvious avenues for analysis. For example,
the activations in a network of this sort would be extremely sparse,
so mathematics from the study of sparseness might be brought to bear.
Other practical extensions also come immediately to mind.

Line 141, grammar, "only subset"
Summary: Adds winner-take-all modules to a feedforward architecture, and achieves significant performance improvements.

Submitted by Assigned_Reviewer_2

This paper introduces a simple form of nonlinearity into neural net architectures - forming groups of (typically two) neurons and zeroing out all the neurons in a group except the one with the highest value. While a very simple idea, it seems to give good results on classification. The authors also give evidence that the network does not forget as much as networks with more standard nonlnearities, however there might be a problem with that experiment (see below).

This paper introduces a simple form of nonlinearity into neural net architectures - forming groups of (typically two) neurons and zeroing out all the neurons in a group except the one with the highest value. While a very simple idea, it seems to give good results on classification. The authors also give evidence that the network does not forget as much as networks with more standard nonlnearities, however there might be a problem with that experiment (see below).

1) There are better results on permutation invariant mnist, see table 1. in http://arxiv.org/pdf/1302.4389v3.pdf .Some of them are just feedforward networks. Also I don't agree that droput in the input should be considered a data augmentation since it doesn't assume anything about the input structure. You should have tried that experiment too.

2) It is a good property for the network not to forget. However the experiments could have few issues. You wait until network reaches certain likelihood and then change the data/labels. Since the new nonlinearity peforms better on recognition, it doesn't have to work as hard to reach the likelihood and so it doesn't need to do so much training, and so the reason it doesn't forget as much can simply be that it didn't train as much to forget what it has learned.

3) You should have also done a similar and related experiment. You have obtained the perofmance when you train on all digits at the same time (till convergence). Now, train till convergence one digits 1,2,3,4,5, then add the remaining digits and train on all digits. This tests how much is the network stuck in the minimum it found when training in the first phase. Ideally the performance in the second experiment (training on 1-5 and then training on 1-10) is the same as training on 1-10 from the start. It is known that this is not the case for sigmoid networks.

Quality: Good, but more experiments should be there and the forgeting expriment should be better.
Clarity: Very good
Originality: I haven't seen it before. On negative side it is just another nonlinearity, related to max pooling, on the positive side it is a simple idea the gives good result.
Significance: Somewhat significant - another nonlinearity into neural networks toolbox.
Summary: It is a simple idea that seems to work well for classification. It is also important for network not to forget, however I think the experiment presented there is not quite correct. Few other experiments would also be useful.

Submitted by Assigned_Reviewer_7

This paper presents a local winner take all approach, where units are grouped in small sets (2 in most of the paper) and only the max of them gets to output something, all the other ones output 0. Experiments on MNIST and a sentiment analysis task show improvements relative to other approaches.

There are connections to max-pooling (only one unit out of a subset fires) and dropout (some units are shut down), but it is indeed different from both and provides a new nonlinearity to consider in the deep learning toolbox.

Results on MNIST are not very impressive. Either the approach gets 0.02% better than the competitor, which means having correctly classified 2 more images out of 10000, or it is simply not as good as normal CNN (but again, by only 0.02%...). [I must say I stopped being impressed by any results on MNIST long time ago...]. The only good result is on the amazon sentiment analysis task, it seems.

I also liked the experiment of section 6, but wondered if it could be shown to improve performance on a real task instead of this artificial setting.

I would have liked the authors to experiment with variying the "pool" size and try to understand when and how it would help: is it better when the model is overfitting, underfitting, noisy, etc.
Summary: A local winner take all technique is described for deep learning algorithms. It provides yet another simple non-linearity to consider in the toolbox. Results on MNIST are not very impressive, but results on sentiment analysis are.
Author Feedback

Author rebuttal: Re: Reviewer_2, point 3: An interesting suggestion. We are
investigating a similar issue in a followup study, but note that this
will need comparisons not just of performance on full set but also of
required training times etc, since if only performance is the issue,
one may increase the learning rate sufficiently after adding the new
digits to get out of the local minimum (later reducing it again).

Re: Reviewer_7: To focus on the effectiveness of the nonlinearity and not
just test set scores on MNIST, we did not add any tricks such as
dropout, unsupervised pre-training, sparsity, etc. When comparing to
other activation functions our result of 1.28% error is markedly better than
the best reported value of 1.43% for rectified linear units (for some
reason, this figure is not cited much) and 1.60% for sigmoidal
units. Though it is not fair to compare to a model with dropout,
(since dropout averages many models implicitly) we think it is
interesting that it is comparable (better by 0.02%) to training a
model with dropout in the hidden layers even though no model averaging has
been utilised.To make this fact clear, we will move the dropout result from
the comparison table and clarify in the main text.

We plan to do further experiments in line with the forgetting
experiment in Section 6, and improve this section as much as the space
constraints allow.